# Weight Recidivism and Dumping Syndrome after Roux-En-Y Gastric Bypass: Exploring the Therapeutic Role of Transoral Outlet Reduction

**DOI:** 10.3390/jpm12101664

**Published:** 2022-10-06

**Authors:** Maria Valeria Matteo, Camilla Gallo, Valerio Pontecorvi, Vincenzo Bove, Martina De Siena, Giorgio Carlino, Guido Costamagna, Ivo Boškoski

**Affiliations:** 1Digestive Endoscopy Unit, Fondazione Policlinico Universitario Agostino Gemelli IRCCS, 00168 Roma, Italy; 2Centre for Endoscopic Research Therapeutics and Training (CERTT), Università Cattolica del Sacro Cuore, 00168 Roma, Italy; 3Division of Gastroenterology and Center for Autoimmune Liver Disease, Ospedale San Gerardo, Department of Medicine and Surgery, University of Bicocca, 20900 Monza, Italy; 4Gastroenterology Unit, Department of Life, Health and Environmental Sciences, University of L’Aquila, 67100 L’Aquila, Italy

**Keywords:** obesity, gastric bypass, weight regain, dumping syndrome, transoral outlet reduction, bariatric endoscopy

## Abstract

Obesity is a chronic, relapsing disease representing a global epidemic. To date, bariatric surgery is the most effective treatment for morbid obesity in the long-term. Roux-en-Y gastric bypass (RYGB) is one of the most performed bariatric interventions, with excellent long-term outcomes. However, about one-third of patients may experience weight regain over time, as well as dumping syndrome. Both these conditions are challenging to manage and require a multidisciplinary and personalized approach. The dilation of the gastro-jejunal anastomosis is a recognized etiological factor for both weight regain and dumping syndrome. Dietary modifications, behavioral interventions, and medications represent the first therapeutic step. Revisional surgery is the traditional approach when non-invasive treatments fail. However, re-interventions may be technically difficult and are associated with increased morbidity and mortality. Transoral outlet reduction (TORe) is an endoscopic procedure aimed at reducing the size of the anastomosis and is proposed as a minimally invasive treatment of weight regain and/or dumping syndrome refractory to conservative therapies. This review is aimed at providing a narrative overview of the role of TORe as part of the multidisciplinary therapeutic toolkit nowadays available to approach weight regain and dumping syndrome after RYGB.

## 1. Introduction

Obesity is a chronic multifactorial condition that has recently become a global epidemic [1]. It represents a challenging condition to manage, requiring a multidisciplinary and personalized approach [2]. Currently, bariatric surgery still represents the most effective therapy for morbid obesity [3,4,5]. Roux-en-Y gastric bypass (RYGB) is one of the most performed bariatric interventions, with excellent long-term results as concerns weight loss and comorbidities improvement [6,7]. Despite that, about one-third of patients may experience weight regain over time [8,9], as well as the onset of long-term complications, including dumping syndrome [10]. 

Weight recidivism after RYGB is an extremely relevant issue since it may lead to comorbidities recurrence and deterioration in the quality of life [2]. This event has been associated with several factors including behavioral, psychological, and anatomical factors, such as the dilation of the gastro-jejunal anastomosis leading to a faster gastric emptying and a reduced sense of satiety [2,11,12].

The enlargement of the gastro-jejunal anastomosis (GJA) is also associated with the incidence of dumping syndrome, which has been reported to arise in 20–50% of patients after RYGB [13,14]. Dumping syndrome includes a set of symptoms related to a series of pathophysiological events triggered by the rapid passage of undigested food into the small bowel [15]. Dumping syndrome is classified as early or late based on pathophysiology and the onset of clinical manifestations [15]. This clinical condition can be physically and mentally debilitant, with a significant impact on everyday life [10,16]. Symptoms detection is pivotal for the diagnosis of this condition [10]. The most used symptoms-based questionnaire for a diagnosis and severity assessment of dumping symptoms is Sigstad’s dumping score [17]. This score assigns points to 16 symptoms of dumping elicited by a high-carbohydrate meal. A Sigstad’s dumping score of ≥7 is strongly suggestive of dumping syndrome, whereas a score of <7 suggests considering another diagnosis. In the case of dumping syndrome, a non-invasive approach represents the first therapeutic step; it includes dietary modifications, dietary supplements that increase the viscosity of food, and medications [10,15,18,19,20,21,22,23].

With referral to interventional approaches, revisional surgery for weight regain and/or intractable dumping syndrome can be technically challenging due to modified anatomy and adhesions, and compared to first-time surgery, it is associated with an increased risk of complications, morbidity, and mortality [24,25]. Furthermore, outcomes of re-intervention for dumping syndrome are controversial [10,26,27,28].

In recent years, endoscopic techniques for the reduction of a dilated GJA, such as sclerotherapy and superficial or full-thickness suturing devices, have been proposed as minimally invasive and repeatable therapeutic options [29,30]. Among them, the transoral outlet reduction (TORe) is currently widely performed; it uses a full-thickness suturing device (Apollo OverstitchTM, Apollo Endosurgery, Austin, TX, USA) combined with argon plasma coagulation (APC) [24,29]. The procedure consists of the cauterization of the anastomotic rim with APC, followed by endoscopic suturing; the use of APC is aimed at strengthening the attachment of the mucosa to ensure a stronger and greater reduction of the anastomotic rim [31,32] (Figure 1). This technique proved better results compared to previously promulgated superficial suturing devices, as it provides greater durability of the restriction of the anastomosis [24,29]. 

The aim of this review is to provide a narrative overview of the role of TORe, which plays a pivotal role in the complete therapeutic toolkit nowadays available to approach challenging obesity complications, mainly referring to weight regain and dumping syndrome. 

## 2. TORe for Weight Regain

Since its first description in 2013 [31], TORe with full-thickness suturing has been applied to several interventional studies in order to prove its effectiveness in different fields of application. TORe’s initially most widespread scope was the management of weight regain after bariatric surgery. The first published study by Jirapinyo et al. included 25 patients with a mean weight regain of 24 kg (37.9%) after RYGB; after undergoing TORe, the included patients registered a mean weight loss of 11.5 kg, 11.7 kg, and 10.8 kg at 3, 6, and 12 months respectively [31]. 

Vargas et al. performed a multi-center retrospective analysis including 130 patients (with a mean weight regain of 24 kg) [33]. Each patient underwent a multidisciplinary assessment, including an evaluation for behavioral interventions before TORe; in particular, a healthy lifestyle change program was encouraged. This study showed an absolute weight loss of 9.31 ± 6.7 kg at 6 months, 7.75 ± 8.4 kg at 12 months, and 8 ± 8.8 kg at 18–24 months. Furthermore, 67% of the patients achieved a total body weight loss (TBWL) of >5% at 1 year. Exceeding this threshold of weight loss is clinically relevant since it has been related to the improvement of multiple obesity-related comorbidities, such as diabetes, hyperlipidemia, and arterial hypertension [34,35]. The authors confirmed the reproducibility of their results through a meta-analysis including two other previous studies, with a total of 330 patients (26; 40): the pooled absolute weight loss was 9.5 kg (95% CI 7.9–11.1), 8.4 kg (95% CI 6.5–10.3), and 8.4 kg (95% CI 5.9–10.9) at 6, 12, and 12–18 months, respectively, with no significant heterogeneity across the included studies (*p* = 0.07). 

In a retrospective study by Tsai et al., 81 patients (with a mean weight regain of 19.2 kg) showed an absolute weight loss of 8 kg at 12 months after TORe [34]. Better weight loss results were associated with the use of two sutures compared to one suture, and a higher BMI at the baseline. In this series, 16 (19.9%) patients repeated TORe within 1 year to achieve an adequate reduction of the anastomosis, and 11 (13.6%) patients required a laparoscopic pouch revision. 

A more recent meta-analysis by Dhindsa et al. including 850 patients from 13 independent cohorts showed an absolute weight loss of 6.14 kg, 10.15 kg, and 7.14 kg at 3, 6, and 12 months, respectively. The mean pooled percentage of the TBWL was 6.69% at 3 months, 11.34% at 6 months, and 8.55% at 12 months [29].

As regards long-term outcomes, a prospective series of 150 patients by Kumar et al. showed an absolute weight loss of 10.5 kg at 1 year and 9.5 ± 2.1 kg at 3-year follow-up [24]. The authors emphasize that TORe was effective in halting weight regain and facilitating permanent weight loss, with a low number needed to treat (NNT); in more detail, the NNT to arrest weight regain was 1.0 at 6 months, 1.1 at 1 year, and 1.2 at 2 and 3 years, while the NNT to maintain weight loss above 5 kg was 1.2, 1.5, 1.9, and 2.0 at 6 months, and 1, 2, and 3 years, respectively. 

Five-year outcomes of TORe have been reported by Callahan et al. in a retrospective analysis of 70 patients with a mean weight regain of 27.5 ± 32.1 kg (42.8 ± 18.7%) after RYGB [36]. This study showed a weight loss of 8.5 ± 11.5 kg at 1 year (n = 42/70), 5.3 ± 9.1 kg at 3 years (n = 31/70), and 3.9 ± 13.1 kg at 5 years (n = 18/70) following TORe. To note, only 26% of patients reached the 5-year follow-up. 

Better 5-year outcomes have been described in a retrospective cohort of 331 patients by Jirapinyo et al. [35]. This study reported percentages of a TBWL of 8.5% at 1 year (n = 276/331 patients), 6.9% at 3 years (n = 211/331), and 8.8% at 5 years (n = 102/331) after TORe. The mean absolute weight loss reported was 9.4 ± 12.3 kg, 8.7 ± 13.8 kg, and 10.3 ± 14.6 kg at 1, 3, and 5 years, respectively. Moreover, almost all the patients had a cessation of weight gain (NNT 1.3), and 62% were able to maintain a TBWL of > 5% at 5 years. On both the univariable and multivariable linear regression analyses, the weight loss during the first year was a predictor of TBWL at 5 years. Of the three-hundred and thirty-one patients, ninety-five (28.7%) underwent an additional endoscopic procedure, sixty-two (18.7%) required weight-loss medications, and four (1.2%) underwent a surgical revision for weight regain, with the necessity of combined treatment in some patients. 

As the diameter of the GJA has a positive linear association with weight regain, the specific goal of the TORe technique is to obtain a durable anastomotic rim restriction [29,37]. For this purpose, several suture patterns (i.e., interrupted, running, single, and double purse-string sutures) have been tried, though, according to updated published results, no recognized gold standard technique has been identified. However, some data suggest that the use of a purse-string suture pattern is associated with a greater weight loss [36,38]; in a comparative study by Schulman et al., in fact, the purse-string pattern proved better weight loss outcomes at 1 year compared with the traditional interrupted method [39].

Furthermore, a modified TORe technique for weight loss has recently been described: it combines full-thickness suturing with anastomotic rim endoscopic submucosal dissection (ESD). The rationale of the dissection before the suturing is to enhance the scarring process and, thus, the durability of the anastomotic restriction [40]. A study comparing traditional TORe and modified ESD-TORe reported a greater reduction in the anastomotic diameter in the second group at 3 months, but no significant difference in terms of weight loss between the two groups at 12 months [40]. Conversely, another study reported greater weight loss at 12 months in the ESD-TORe group compared to patients undergoing traditional TORe with APC [41]. However, ESD is technically more demanding than APC, and it may increase the risk of procedure-related major complications, mainly perforation and bleeding [42]. This may limit the spreading of this technique in clinical practice.

As reported in Table 1, considering the most statistically reliable studies regarding TORe for weight regain, the overall data show a mean time between the RYGB and TORe of 7.8 ± 1.2 years, and a mean GJA diameter reduction of 18.5 ± 3.7, with a mean absolute weight loss of 9.6% ± 2.2 at 6 months, of 9.1% ± 1.3 at 12 months, of 8.5% ± 0.7 at 2 years, of 7.8% ± 2.2 at 3 years, and of 6.65% ± 4.5 at 5 years.

With reference to procedural safety, traditional TORe proved to have a good safety profile. According to a recent meta-analysis [29] including 850 patients and 877 TORe procedures, in fact, the reported pooled rate of severe adverse events, such as bleeding (melena or hematemesis), anastomotic stenosis, and perforation, was 0.57% ± 1.35%. The most frequently complained about post-operative symptom was abdominal pain, with a pooled incidence rate of 4.22 ± 8%, which has usually been managed with symptomatic drugs. To date, no fatal adverse events after TORe have been reported.

## 3. TORe for Dumping Syndrome

Although most of the available evidence refers to the treatment of weight regain after RYGB, some studies evaluating TORe for dumping syndrome have been recently published. 

In a small series of 14 patients with late dumping syndrome, Stier et al. reported a significant reduction of Sigstad’s dumping score (12.71 ± 4.18 vs. 3.07 ± 2.06; *p* < 0.001) 4 weeks after TORe [44]. Patients underwent post-procedure scintigraphy showing a considerably delayed gastric emptying compared to the baseline scintigraphy. Only one patient had persistent symptoms and required surgical revision. 

In a retrospective cohort of 40 patients with dumping syndrome, Tsai et al. showed a significant decrease in Sigstad’s dumping score from 13.9 (0–28) at the baseline to 8.6 (0–28) at a mean follow-up of 14.8 (3–32) months after the procedure [45]. Nine patients repeated the endoscopic procedure for persistent/recurrent symptoms. Of these, seven patients had remission, while the other two required a laparoscopic revision.

Vargas et al. published a multicenter prospective study including 115 patients with dumping syndrome, defined as a Sigstad’s score of ≥ 7 and non-responsive to medical therapy [46]. After TORe, the mean Sigdtad’s score dropped from 17.23 ± 5.9 to 2.55 ± 1.87 at the 3-month follow-up (*p* < 0.0001). They reported nine (3%) failures with a recurrence of dumping syndrome; of these, three patients repeated TORe because of a GJA dilation found at a repeated endoscopy, and three underwent an enteral feeding tube insertion. Parallelly, this interventional study proved an effective TORe-related weight loss (with a mean TBWL of 9.3% and mean absolute WL of 9.3 kg at 3 months). 

Relly et al. reported an 85% (11/13) resolution rate of dumping syndrome (Sigstad’s score of < 7) at 6 months after TORe, with a reduction of Sigstad’s dumping score from 19.4 to 5.2 (*p* < 0.001) [32]. Two patients underwent a second TORe because of inadequate clinical response and endoscopic evidence of anastomosis dilation (>1 cm) with an enlargement of the suture line. 

As regards long-term results, a retrospective study by Brown et al. showed the resolution of dumping syndrome in 80% of patients at 2 years after an endoscopic revision [47]. The authors mention that the patients were followed-up by a multi-disciplinary medical and surgical weight loss team. 

Furthermore, in a recent large study by Petchers et al. including 98 patients, the reported resolution rate of dumping syndrome was 88% at one month and 84% at the long-term follow-up after TORe (an average of 3.5 years) [48]. In this series, only 7% of the included patients required a second intervention 2–3 years after the first one because of symptoms recurrence and endoscopic evidence of a recurrent enlargement of the anastomosis. 

As for TORe for weight regain, different suture patterns have been proposed also for dumping syndrome treatment, with no evidence supporting the superiority of a particular suturing technique. 

Table 2 summarizes the main available data on TORe for dumping syndrome. The median time between RYGB and TORe is 6.9 ± 1.9 years; the reported mean reduction in the diameter of the GJA is 21.8 ± 10.5 mm, but no reliable data can be extracted regarding the clinical follow-up and, thus, the improvement of dumping syndrome, given the fact that the considered studies applied different methods to assess the symptoms and observed the patients at different follow-up times. 

TORe’s good safety profile was confirmed also with regards to dumping syndrome treatment, as no severe adverse events have been reported, except for Petchers et al. who described acute gastrointestinal bleeding from anastomotic ulceration within 30-days after the procedure [32,44,45,46,47,48].

## 4. Discussion

Roux-en-Y gastric bypass has proved excellent long-term results in the treatment of morbid obesity. Nevertheless, weight recidivism may occur over time in about 30% of patients, with a recurrence of obesity-related comorbidities and deterioration in the quality of life [8,9,49,50]. The etiology of weight regain is multifactorial, as it includes psychological factors, endocrine imbalances, nutritional non-compliance, physical inactivity, and follow-up loss alongside anatomical remodeling, such as gastro-enteric anastomosis widening [2]. 

As such, the management of weight regain is challenging, requiring a dynamic and personalized approach to the patients, guided by a multidisciplinary team [2]. The cooperation between endocrinologists, psychologists/psychiatrists, nutritionists, physical educators, endoscopists, and surgeons is, in fact, crucial to building the best therapeutic strategy for each patient [51].

At first, the correction of any psychological, endocrinological, and nutritional issues is mandatory [2]. However, the influence of anatomic alterations cannot be overlooked, and an endoscopic evaluation of the anastomosis should be offered in every case of significant weight regain. 

TORe represents an endoscopic procedure aimed at narrowing the GJA to delay gastric emptying and to induce a prolonged satiety. This non-invasive repeatable procedure proved to be safe and effective in several studies, with initial evidence of efficacy up to 5 years [24,29,30,31,33,35,36,43]. Notably, according to latest reliable interventional studies, about two thirds of patients undergoing TORe were able to maintain a TBWL >5% at 1- and 5-years follow-up [33,35]. As already mentioned, this result is of special relevance since a total weight loss above 5% is associated with significant comorbidities improvement [33,34,35]. 

As such, TORe represents a low-risk treatment that should be included in the multidisciplinary decision-making process for weight regain after RYGB. 

Similarly, dumping syndrome represents another common complication after RYGB [10]. This condition is characterized by multiple clinical manifestations elicited by the rapid movement of ingested food from the stomach into the small bowel [15]. The enlargement of the anastomosis plays a key role in the etiology of dumping syndrome since it can significantly accelerate this passage [43]. Early dumping symptoms, including abdominal cramps, nausea, tachycardia, and diarrhea, usually arise within 30–60 min after a meal; it is caused by the osmotic fluid shift from vessels to the jejunal lumen, which results in a reduction of the circulating blood volume, intestinal distention, and release of gastrointestinal peptide hormones [10,15]. Late dumping syndrome is characterized by hypoglycemic manifestations arising 1–3 h after a meal and is supposed to be induced by the rapid absorption of carbohydrates that boosts insulin release [10,15].

Although avoiding food ingestion to prevent dumping symptoms may conceptually increase weight loss, some patients suffer from particularly disabling symptoms with a significant impact on their quality of life [10].

Dietary adjustments represent the first therapeutic step in the management of dumping syndrome; it is specifically recommended to reduce the quantity of food consumed at each meal, delay fluid intake until at least 30 min after meals, avoid simple sugars, and increase fiber-rich food intake [10,15]. When dietary modifications fail, dietary supplements that increase the viscosity of food and medications, such as acarbose and somatostatin analogues, may be introduced [10,18,19,20,21,22,23]. However, the adherence of patients to these therapies may not be adequate due to adverse effects (such as flatulence and swelling in the case of acarbose administration, and diarrhea in the case of octreotide), high costs, and the mode of their administration (somatostatin analogues are injected subcutaneously) [10]. 

Surgical interventions, including stomal revision, pyloric reconstruction, Billroth II to Billroth I anastomoses, jejunal interposition, and Roux-en-Y conversion and enteral nutrition, represent the traditional options for patients with refractory symptoms [52]. However, re-interventions are characterized by increased surgical and post-interventional risks and did not prove certain effectiveness in terms of a resolution of post-bariatric surgical complications [10,26,27,28].

A large systematic review by Tran et al. evaluated the outcomes of several methods of surgical revision after RYGB, including conversion to distal Roux-en-Y gastric bypass (DRYGB), revision of the gastric pouch and anastomosis, revision with a gastric band, and conversion to a biliopancreatic diversion/duodenal switch (BPD/DS) [28,53,54,55,56,57,58,59,60,61,62,63]. The weighted averages of major complication rates (i.e., leak, significant bleeding, acute abdomen, band migration, abdominal abscess, and severe malnutrition) were 11.9% for DRYGB, 4% for BPD/ DS, 3.8% for gastric banding revision, and 3.5% for pouch/stoma revision surgery [28]. To note, these rates far exceed those reported for TORe (<1%). Furthermore, there is no evidence of sustained weight loss for both the gastric band and revision of the gastric pouch and anastomosis [53,54,55,56]. DRYGB has proved to have good long-term results, even though with a major complication rate of 11.9%, half of which is related to severe malnutrition [28,57,58,59,60,61]. A conversion to a biliopancreatic diversion/duodenal switch (BPD/DS) has proved excellent long-term weight loss outcomes, with a complication rate of 4%. However, this surgery is performed only in a few centers because of its technical complexity, thus limiting its use on a large scale [28,62,63].

Endoscopic transoral outlet reduction (TORe) has recently been proposed as a minimally invasive treatment for patients with dumping syndrome refractory to medical therapy after RYGB [32,44,45,46,47,48]. The rationale for this procedure is to reduce the diameter of the anastomosis, thus delaying gastric emptying [32,44]. Some recent studies described the efficacy of this technique in this clinical scenario. A statistically significant reduction of Sigstad’s dumping score has been reported in four studies at different times during follow-ups within 1 year after TORe [32,44,45,46]. In addition, two studies showed an 80% and 84% resolution of dumping syndrome at 2- and 3.5-year follow-ups, respectively [47,48]. However, the enlargement of the anastomosis may not be the only etiologic factor involved in dumpling syndrome, and this is probably the reason why a minority of patients do not respond to anastomosis reduction [46]. Larger prospective studies are certainly needed to further confirm the role of TORe in the treatment of dumping syndrome, but to date, available data are encouraging. 

TORe demonstrated having a satisfactory safety profile, with a < 1% rate of severe adverse events and no procedure-related deaths reported [29,30,31,32,33,35,36,43,44,45,46,47,48].

Overall, given the advantageous safety profile and the promising efficacy results, TORe can be considered the first minimally invasive interventional step in the case of weight regain and/or dumping syndrome refractory to medical and behavioral treatments (Figure 2). 

Furthermore, TORe is repeatable, per definition, as all endoscopic procedures are. Some authors reported the successful use of 1–2 additional TORe for both weight regain and dumping syndrome [32,33,35,45,46,48]. To note, only a minority of patients with an inadequate response to the first TORe underwent surgical revision (see Table 1 and Table 2). As such, the necessity of a redo-TORe should not be judged as a treatment failure, but as part of the overall endoscopic approach. However, each case should be revised and discussed by the multidisciplinary team before giving the indication of a redo-TORe, as for any case of first post-bariatric surgery complications. Given the fact that both weight relapse and dumping syndrome are complex conditions, patient selection for an endoscopic approach and regular follow-up should be always entrusted to an experienced multidisciplinary team to early modify the therapeutic strategy if necessary. As a matter of fact, a tailored and dynamic strategy is crucial in defining the long-term effectiveness of the management of each bariatric patient. 

Given the currently limited evidence on TORe for dumping syndrome, further studies investigating this minimally invasive technique with a larger sample size and long-term follow-up are needed. With referral to the comprehensive therapeutic strategy for both weight regain and dumping syndrome after RYGB, the role of the ancillary multidisciplinary team, including nutritionists, endocrinologists, and psychologists, and TORe should also be investigated by including an integrated therapeutic program in the study protocols. The evaluation of adherence to a multidisciplinary follow-up and its impact on the efficacy of the outcomes also represents an unmet need. Overall, randomized head-to-head studies comparing the efficacy for both weight regain and dumping syndrome and the safety of surgical revision and TORe would be of high scientific value. However, the practical feasibility of such studies may be limited by the need for a large sample size and homogeneous cohorts and by the heterogeneity of surgical and endoscopic techniques among different centers.

## 5. Conclusions

TORe is a minimally invasive procedure that has proven to be safe and effective in inducing weight loss in patients with weight regain and, more recently, in the treatment of dumping syndrome after RYGB. As with every bariatric procedure, TORe cannot be prescinded from long-term management by a bariatric multidisciplinary team operating on different levels, including dietary and behavioral habits. The therapeutic strategy in such complex conditions should be tailored to each patient and requires dynamic interactions between the several health professionals of the bariatric multidisciplinary team. A multimodal tailored strategy that combines nutritional, psychological, medical, and endoscopic interventions is probably the best option, while revisional surgery should be reserved for patients refractory to conservative and minimally invasive treatments.

## Figures and Tables

**Figure 1 jpm-12-01664-f001:**
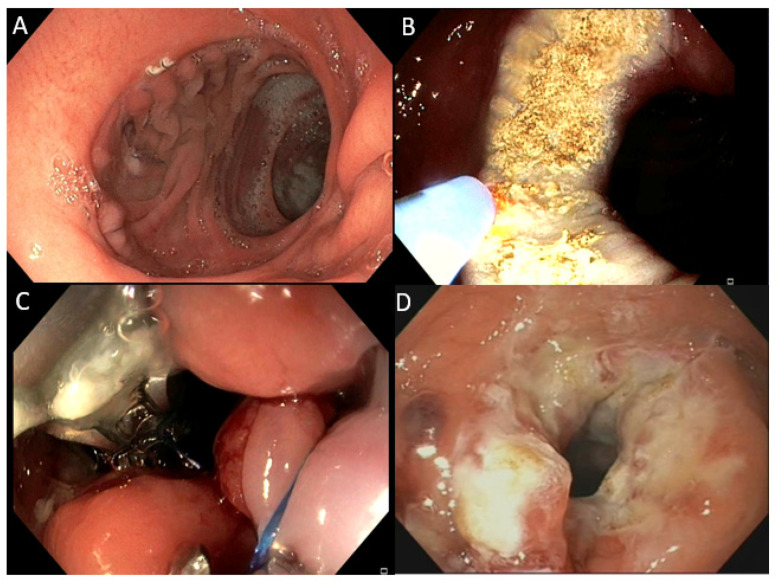
Transoral outlet reduction of the GJA in RYGB: (**A**) enlarged GJA anastomosis; (**B**) cauterization with APC; (**C**) suturing with Apollo Overstitch; (**D**) final reduction.

**Figure 2 jpm-12-01664-f002:**
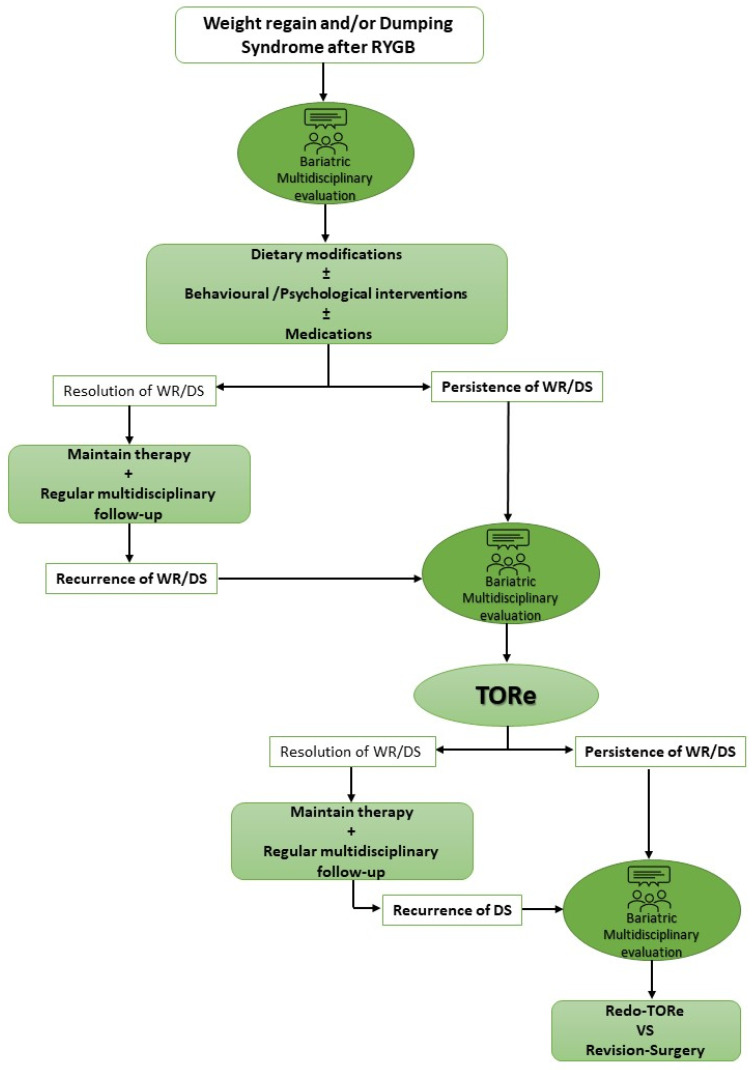
Proposed flow-chart for weight regain and/or dumping syndrome management after RYGB. WR: weight regain; DS: dumping syndrome; RYGB: Roux-en-Y gastric bypass; EW: excess weight; TORe: transoral outlet reduction.

**Table 1 jpm-12-01664-t001:** Summary of the main studies evaluating traditional full-thickness TORe (APC-Apollo Overstitch) for weight regain after gastric bypass.

Study	N of Patients	Time RYGB–TORe (Years)	Pre-TORe Weight Parameters	Weight Regain after RYGB	Pre-TORe GJA Diameter (mm)	Post-TORe GJA Diameter (mm)	Weight Loss 6m	Weight Loss 12m	Weight Loss 2y	Weight Loss 3y	Weight Loss 5y	Redo-TORe	Surgery
Jirapinyo et al. [31]	25	6 (2–10)	BMI: 43 kg/m^2^	24 kg (1.4–59) (37.9%) **	26.4 (18–40)	6 (3–10)	11.7 kg (2.3–27.2)	10.8 kg (0.7–27.2)	NA	NA	NA	NA	NA
Kumar et al. [24]	150	8.6 ± 0.3	Weight: 110.7 ± 2.2 kg BMI: 40.1 ± 0.7 kg/m^2^	4.1 ± 0.3 kg (49.7 ± 4.3%)	24.1 ± 0.6	9.0 ± 0.2	AWL: 10.6 ± 0.7 kg TBWL: 9.6 ± 0.6% EWL: 28.8 ± 2.7%	AWL: 10.5 ± 1.2 kg TBWL: 9.5 ± 0.9% EWL: 24.9 ± 2.6%	AWL: 9.0 ± 1.7 kg TBWL: 8.1 ± 1.4% EWL: 20.0 ± 6.4%	AWL: 9.5 ± 2.1 kg TBWL: 8.6 ± 1.5% EWL: 19.2 ± 4.6%	NA	NA	NA
Vargas et al. [33]	130	8.4 ± 4.78	BMI: 36.8 ± 6.84 kg/m^2^	24.6 ± 16.6 kg (38.8%)	28 ± 4.74	8.3 ± 1.42	AWL: 9.31 ± 6.7 kg	AWL: 7.75 ± 8.4 kg TBW: 6 ± 7.0% EWL: 20.2 ± 10%	AWL: 8 ± 8.8 kg*	NA	NA	NA	NA
Tsai et al. [43]	81	6.7 (0.8–18.5)	Weight: 94.9 kg BMI: 33.6 kg/m^2^	18.2 kg (36.1%) **	22 (13–40)	6 (4–14)	AWL: 6.0 (0.2–24.8) kg	AWL: 8.0 (0.2–8)	NA	NA	NA	16 (19.9%)	11 (13.65%) lap pouch revision
Jirapinyo et al. [35]	331	9.3 ± 4.7	Weight: 110.0 ± 26.3 kg BMI: 40.1 ± 9.1 kg/m^2^	55.2 kg (51.0%) **	23.4 ± 6.0	8.4 ± 1.6	NA	AWL: 9.4 ± 12.3 kg TBWL: 8.5 ± 8.5%	NA	AWL: 8.7 ± 13.8 kg TBWL: 6.9 ± 10.1%	AWL: 10.3 ± 14.6 kg TBWL 8.8 ± 12.5%	95 (28.7%)	4 (1.2%) GJA reconstruction or limb distalization
Callahan et al. [36]	70	7.7 ± 4.0	Weight: 116.1 ± 25.2 kg BMI: 42.3 ± 8.5 kg/m^2^	27.5 ± 32.1 kg (42.8 ± 18.7%)	30.6 ± 6.2	5.8 ± 2.0	AWL: 10.7 ± 11.6 kg EWL: 18.5 ± 18.2%	AWL: 8.5 ± 11.5 kg EWL: 14.9 ± 20.6%	NA	AWL: 5.3 ± 9.1 kg EWL: 8.7 ± 14.9%	AWL: 3.9 ± 13.1 kg EWL: 7.0 ± 23.8%	NA	NA

AWL: absolute weight loss; BMI: body mass index; EWL: excess weight loss; GJA: gastro-jejunal anastomosis; NA: not available; TBWL: total body weight loss. Values are reported as mean values ± standard deviation or (range) as reported in each study. * 18–24 months; ** when non-specified by authors, absolute weight regain and/or percentage has been calculated based on absolute mean values of weight after Roux-en-Y Gastric Bypass, at nadir, and before TORe.

**Table 2 jpm-12-01664-t002:** Summary of the main studies evaluating traditional full-thickness TORe (APC-Apollo Overstitch) for dumping syndrome after gastric bypass.

Study	N. of Patients	Time RYGB–TORe (Years)	Pre-TORe GJA Diameter (mm)	Post-TORe GJA Diameter (mm)	SDS Baseline	Dumping Syndrome Outcomes	Redo-TORe	Surgery
Stier et al. [44]	14	4.6 ± 2.6	NA	8	12.7 ± 4.2	1 month: SDS 3.1 ± 2.1	0/14 (0%)	1 (reconstruction of the upper gastrointestinal tract + sleeve gastrectomy)
Brown et al. [47]	27	NA	NA	12	NA	3 months: 92% DS resolution 2 years: 80% DS resolution	NA	NA
Tsai et al. [45]	40	6.7 (0.8–19.0)	22.6 (18–35)	< 10	13.9 (0–28)	14.8 (3–32) months: SDS 8.6 (0–28)	9/40 (22.5%)	2 (laparoscopic pouch revision)
Vargas et al. [46]	115	8.9 ± 1.1	39.8 ± 6.7	6.2 (4–13)	17.2 ± 5.9	3 months: SDS 2.6 ± 1.9	3/115 (2.6%)	3 (surgical enteral feeding tube placement)
Relly et al. [32]	13	5.5 (1–9)	25.2 (15–30)	5.6 (5–10)	19.4 ± 3.6	6 months: SDS 5.2 ± 5.5	2/13 (15.2%)	NA
Petchers et al. [48]	98	9 ± 4.6	NA	15	NA	1 month: 88% DS resolution 3.5 years: 84% DS resolution	7/98 (7.1%)	NA

AWL: absolute weight loss; BMI: body mass index; DS: dumping syndrome; GJA: gastro-jejunal anastomosis; NA: not available; SDS: Sigstad’s dumping score; TBWL: total body weight loss. Values are reported as mean values ± standard deviation or (range) as reported in each study.

## Data Availability

Not applicable.

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
