# Peer review of "Weight Recidivism and Dumping Syndrome after Roux-En-Y Gastric Bypass: Exploring the Therapeutic Role of Transoral Outlet Reduction"

_jpm, 2022, doi:10.3390/jpm12101664_

Round 1
Reviewer 1 Report
This article more or less review article , there cases did not add any new information for this problem
Author Response
General comment: This article more or less review article, there cases did not add any new information for this problem
Answer: Thank you for your comment. This is a narrative review of the literature. According to the nature and the type of research, this manuscript brings the aspects of the scientific evidence currently available relating to the TORe procedure. In particular, it faithfully photographs the current state of the art, collecting not only the evidence relating to the efficacy of the procedure on weight regain after RYGB, which is more representative, but also the data relating to TORe procedure as a minimally invasive solution for dumping syndrome after RYGB, which are much more limited. This article, therefore, aims to summarize all the achievements of this recent endoscopic technique and underline its scientific limits, thus representing the cornerstone for future perspectives faithfully and punctually. Furthermore, this paper provides insights for a personalized and multidisciplinary approach in these clinical conditions and for further research.
Reviewer 2 Report
The authors produced a narrative review on therapeutic role of Transoral Outlet Reduction in the treatment of obesity complications after Roux-en-Y gastric Bypass, mainly referring to weight regain and dumping syndrome. The manuscript is globally well composed, the narration is clear, the content informative and, globally, it adds some interesting insights into these matters.
Here are some suggestions:
1. Please explain more the limitations of surgery re-interventions with some literature references.
2. I would add some specific future perspectives: e.g., given some of the limitations of the current literature, which kind of studies do we need in the future to improve our understanding of the topic.
Author Response
General comment: The authors produced a narrative review on therapeutic role of Transoral Outlet Reduction in the treatment of obesity complications after Roux-en-Y gastric Bypass, mainly referring to weight regain and dumping syndrome. The manuscript is globally well composed, the narration is clear, the content informative and, globally, it adds some interesting insights into these matters.
Answer: Thank you very much.
Here are some suggestions:
Comment 1: Please explain more the limitations of surgery re-interventions with some literature references.
Answer: Thank you for the precious suggestion. We provided more details about the limitations of revisional surgery in the discussion.
Comment 2: I would add some specific future perspectives: e.g., given some of the limitations of the current literature, which kind of studies do we need in the future to improve our understanding of the topic.
Answer: This is a very good suggestion, thank you. This was added in the discussion.
Reviewer 3 Report
Manuscript ID: JPM
Title: Weight recidivism and dumping syndrome after Roux-en-Y Gastric Bypass: Exploring the therapeutic role of Transoral Outlet Reduction
This paper reviewed an emerging endoscopic technique for the treatment of weight regain and dumping syndrome after RYGB. Although TORe is not a very common procedure, this review is precious. It would be better to compare it with other surgical treatments, such as gastric pouch resizing or anastomosis distalization, but it does not seem easy. This review appears worthy of publication in this journal.
Author Response
Comment: This paper reviewed an emerging endoscopic technique for the treatment of weight regain and dumping syndrome after RYGB. Although TORe is not a very common procedure, this review is precious. It would be better to compare it with other surgical treatments, such as gastric pouch resizing or anastomosis distalization, but it does not seem easy. This review appears worthy of publication in this journal.
Answer: Thank you very much for your comment. To date, there are no studies comparing surgical and endoluminal techniques. We mentioned in the revised discussion that this could be the aim of future prospective randomized trials.